# Preoperative Immune Cell Dysregulation Accompanies Ovarian Cancer Patients into the Postoperative Period

**DOI:** 10.3390/ijms25137087

**Published:** 2024-06-28

**Authors:** Jonas Ulevicius, Aldona Jasukaitiene, Arenida Bartkeviciene, Zilvinas Dambrauskas, Antanas Gulbinas, Daiva Urboniene, Saulius Paskauskas

**Affiliations:** 1Laboratory of Surgical Gastroenterology, Institute for Digestive Research, Medical Academy, Lithuanian University of Health Sciences, A. Mickeviciaus g. 9, LT-44307 Kaunas, Lithuania; aldona.jasukaitiene@lsmu.lt (A.J.); arenida.bartkeviciene@lsmu.lt (A.B.); zilvinas.dambrauskas@lsmu.lt (Z.D.); antanas.gulbinas@lsmu.lt (A.G.); 2Department of Laboratory Medicine, Medical Academy, Lithuanian University of Health Sciences, A. Mickeviciaus g. 9, LT-44307 Kaunas, Lithuania; daiva.urboniene@lsmu.lt; 3Department of Obstetrics and Gynecology, Medical Academy, Lithuanian University of Health Sciences, A. Mickeviciaus g. 9, LT-44307 Kaunas, Lithuania

**Keywords:** ovarian cancer, surgery, peripheral blood mononuclear cells (PBMC), interleukin-1 beta (IL-1β), interleukin-4 (IL-4), interleukin-6 (IL-6), interleukin-10 (IL-10), interleukin-12 (IL-12), tumor necrosis factor alpha (TNFα)

## Abstract

Ovarian cancer (OC) poses a significant global health challenge with high mortality rates, emphasizing the need for improved treatment strategies. The immune system’s role in OC progression and treatment response is increasingly recognized, particularly regarding peripheral blood mononuclear cells (PBMCs) and cytokine production. This study aimed to investigate PBMC subpopulations (T and B lymphocytes, natural killer cells, monocytes) and cytokine production, specifically interleukin-1 beta (IL-1β), interleukin-4 (IL-4), interleukin-6 (IL-6), interleukin-10 (IL-10), interleukin-12 (IL-12), and tumor necrosis factor alpha (TNFα), in monocytes of OC patients both preoperatively and during the early postoperative period. Thirteen OC patients and 23 controls were enrolled. Preoperatively, OC patients exhibited changes in PBMC subpopulations, including decreased cytotoxic T cells, increased M2 monocytes, and the disbalance of monocyte cytokine production. These alterations persisted after surgery with subtle additional changes observed in PBMC subpopulations and cytokine expression in monocytes. Considering the pivotal role of these altered cells and cytokines in OC progression, our findings suggest that OC patients experience an enhanced pro-tumorigenic environment, which persists into the early postoperative period. These findings highlight the impact of surgery on the complex interaction between the immune system and OC progression. Further investigation is needed to clarify the underlying mechanisms during this early postoperative period, which may hold potential for interventions aimed at improving OC management.

## 1. Introduction

Ovarian cancer (OC) is a significant health concern worldwide, with high morbidity and mortality rates. It is recognized as the most fatal gynecologic malignancy, accounting for more than two thirds of mortality within this category [1]. This is largely attributed to the prevalence of advanced-stage ovarian cancer diagnoses in most patients [1,2]. Current treatment modalities, primarily comprising surgery and chemotherapy, notably enhance rates of progression-free survival (PFS) and overall survival (OS), although the long-term prognosis remains unsatisfactory [3]. Researchers continually strive to develop improved treatment strategies, yet the optimal approach for ovarian cancer treatment remains elusive [4,5]. There is a growing recognition of the importance of the immune system in ovarian cancer progression and treatment response with accumulating evidence suggesting that alterations in immune cell populations and cytokine production play crucial roles in the tumor microenvironment and its progression [6,7,8,9].

Peripheral blood mononuclear cells (PBMCs) are a heterogeneous population of immune cells comprising lymphocytes, monocytes, and dendritic cells. PBMCs are pivotal in modulating inflammation and regulating immune responses against tumors [10,11,12]. In OC patients, alterations in PBMC subpopulations and cytokine production have been observed, reflecting the intricate interplay between the immune system and tumor development [12,13]. PBMCs exhibit multifaceted immune responses against cancer through cytotoxicity, cytokine secretion, antigen presentation, and regulatory functions [11]. Functionally diverse mature T lymphocytes (CD3+ T cells) are mainly categorized into CD3+/CD4+ T lymphocytes (CD4+ T cells) and CD3+/CD8+ T lymphocytes (CD8+ T cells), each contributing significantly to antitumor immunity. CD4+ T cells, also known as T helper lymphocytes, modulate immune responses through cytokine secretion, influencing inflammation and tumor growth as well as assisting in the activation of other immune cells [14,15]. Conversely, CD8+ T cells, termed cytotoxic T lymphocytes, eliminate infected or malignant cells via cytokine release and cytolytic molecules [15,16]. B lymphocytes (CD19+ B cells) regulate humoral immunity, influence CD4+ T cell function, and may impact tumor progression, although their specific effect on ovarian cancer remains incompletely understood [17,18]. Innate immune cells, such as CD3−/CD16 + 56+ lymphocytes, also referred to as natural killer (NK) cells, primarily exhibit anticancer activity but constitute a minor fraction of the tumor microenvironment [12,19,20]. Conversely, macrophages, derived from CD14+ cells (monocytes), comprise a substantial portion of the tumor microenvironment and wield significant influence over tumor progression [11,12,21]. The impact of monocytes varies depending on subtype, with differentiation into M1 macrophages possessing tumor-inhibiting properties, while M2 macrophages are characterized by tumor-promoting effects [22]. While these immune cells have defined roles, it is essential to recognize that all PBMCs can be modulated by various factors, potentially altering their effect on cancer progression [11,12]. Furthermore, considering the immune system’s involvement in OC progression, various ratios involving PBMCs and other immune cells, such as the neutrophil to lymphocyte ratio (NLR), platelet to lymphocyte ratio (PLR), lymphocyte to monocyte ratio (LMR), CD4+ T cells to CD8+ T cells (CD4/CD8) ratio, and M1 to M2 monocytes (M1/M2) ratio, are currently under investigation for their prognostic utility in OC patients [23,24,25].

For a more comprehensive understanding of the immune response in OC patients undergoing surgical treatment, it is essential to explore cytokines and their response to surgery. Several crucial cytokines, including but not limited to interleukin-1 beta (IL-1β), interleukin-4 (IL-4), interleukin-6 (IL-6), interleukin-10 (IL-10), interleukin-12 (IL-12), and tumor necrosis factor alpha (TNFα), are pivotal for understanding the complexities of the immune system’s response to cancer [11,26,27]. These cytokines play critical roles in regulating inflammatory reactions, primarily with IL-1β, IL-6, and TNFα being involved. Additionally, they modulate the function and differentiation of immune cells through their immune suppressive effects (IL-4 and IL-10) and immune stimulation effects (IL-12) [11,26].

Cytoreductive surgery is a cornerstone of OC treatment, which is often performed together with chemotherapy. The removal of tumor tissue has long-term benefits by decreasing disease progression, yet short-term tissue damage might counteract these gains. While radical surgery is recognized as an independent positive prognostic factor [28], the subsequent tissue trauma has a significant impact on the immune response through multiple mechanisms, leading to immune dysfunction [29,30], and potentially promoting metastatic progression [31,32]. Despite being substantiated in various cancer types, this phenomenon has not been validated in OC [33,34]. Furthermore, extensive research has investigated alterations in PBMC subpopulations and cell ratios in OC patients, aiming at patient stratification and prognosis assessment [13,23,25,35,36]. However, data regarding surgery’s impact on PBMC subpopulations in OC patients are limited [37,38,39] with scant information available on postoperative cytokine production in PBMCs [40].

We postulated that during the initial postoperative period, OC patients may encounter dysregulation in their peripheral PBMC subpopulations. Our aim was to examine PBMC subpopulations and cytokine production within monocytes following surgery, focusing on the early postoperative period. These results may highlight immune dysregulation during early postoperative recovery in OC patients, contributing to the improvement of OC management strategies.

## 2. Results

### 2.1. Participant Characteristics

The median age of participants in the study group, 58 (49–67) years, was similar to that of the controls. Body mass index (BMI) did not differ significantly between OC patients and controls. The histologic types observed in OC patients reflected those seen in the general population [41]. Further clinicopathologic characteristics of both controls and OC patients are outlined in Table 1.

### 2.2. Alterations in Peripheral Blood Cell Counts, Ratios, and Monocyte Cytokine Expression in Ovarian Cancer Patients

Complete blood count (CBC) analysis was conducted for both controls and OC patients prior to surgery. Our findings indicated a tendency for OC patients to have higher white blood cell (WBC) counts compared to controls, which was primarily attributable to the heightened neutrophil levels in OC patients. Furthermore, OC patients exhibited elevated platelet (Plt) counts alongside a trend suggesting diminished red blood cell (RBC) counts. Analysis of hemoglobin (Hb) concentration indicated that OC patients showed lower levels, measuring 124 (115–130) g/L, compared to controls, which exhibited levels of 135 (126–140) g/L (*p* = 0.01). Detailed data from the CBC analysis are presented in Table 2.

After PBMC isolation, samples underwent analysis to identify alterations in PBMC subpopulations in OC patients. Preoperatively, OC patients exhibited elevated proportions of CD19+ B cells and decreased levels of CD3+ T cells and CD8+ T cell subsets (Figure 1A). Monocyte subpopulation analysis revealed equal proportions of M1 monocytes between controls and OC patients preoperatively. However, M2 monocytes exhibited approximately 2.5 times higher proportions in OC patients preoperatively, while non-classified monocytes demonstrated lower proportions (Figure 1B).

Further analysis of CBC results revealed higher NLR and PLR in the OC group before surgery. Additionally, the analysis of PBMCs ratios showed a significantly higher CD4/CD8 ratio and markedly decreased M1/M2 ratio in OC patients preoperatively. However, the LMR between the investigated groups remained unchanged (Figure 1C). To enhance understanding of the interrelationships among the investigated ratios and all peripheral blood cells, we conducted a detailed correlation analysis. The results of the correlation analysis in preoperative OC patients are illustrated in Figure 2, and those of the control group are presented in Appendix A.

Cytokine expression analysis in the two main subtypes of monocytes revealed that the majority of the analyzed cytokine expressions were comparable between the control group and OC patients with several notable differences observed in M2 monocytes. Specifically, TNFα expression exhibited a tendency to be less expressed, and IL-10 expression was significantly lower in OC patients. Conversely, IL-6 expression in M2 monocytes was higher in OC patients before surgery (Figure 3).

### 2.3. Alterations in Peripheral Blood Cell Counts, Ratios, and Monocyte Cytokine Expression in Ovarian Cancer Patients Postoperatively

The analysis of postoperative changes in PBMC subpopulations in OC patients revealed a spectrum of fluctuations, although the majority of these alterations did not reach statistical significance compared to the preoperative baseline (Figure 4A,B). However, significant differences emerged when postoperative OC patients were compared to controls. One day following surgery, there was a statistically significant increase in CD4+ T cell counts, with values of 36.1 (35.3–37.1)% vs. 40.9 (37.7–46.8)%, respectively (*p* < 0.001). Additionally, the proportion of CD19+ B cells reached its peak 1 day postoperatively, with values of 4.1 (3.8–4.6)% vs. 6.7 (5.7–7.9)%, respectively (*p* < 0.001). Similarly, M2 monocytes exhibited a more than threefold rise in proportion compared to controls with values of 2.3 (1.4–2.8)% vs. 7.5 (6.7–7.8)%, respectively (*p* < 0.001). Conversely, analysis of non-classified monocytes showed opposing results, with levels further decreased 1 day postoperatively at 4.4 (4.1–4.8)% compared to control levels of 9.6 (8.4–12.1)% (*p* < 0.001). The analysis of postoperative PBMC ratios did not reveal significant alterations in OC patients compared to their preoperative state (Figure 4C).

The majority of analyzed cytokine expressions exhibited comparability before and after surgery in OC patients (Figure 5). However, notable differences were observed in comparison to cytokine expression in controls. Specifically, TNFα expression in M1 monocytes reached its peak 1 day postoperatively, being 1.24 times higher compared to controls, with median fluorescence intensity (MFI) values of 634 (449–663) vs. 789 (633–845), respectively (*p* = 0.02). Additionally, following a trend of decreased TNFα expression in M2 monocytes preoperatively, its expression equilibrated to control levels 1 day postoperatively (MFI values 535 (398–683) vs. 487 (298–621), respectively, *p* = 0.64). Similar fluctuations were observed in IL-10 expression in M2 monocytes, with its expression equalizing to controls on day 5 postoperatively (MFI values 640 (426–729) vs. 645 (363–701), respectively, *p* = 0.53). The most pronounced fluctuations were noted in IL-6 expression in M2 monocytes, which peaked on day 1 postoperatively, being 1.5 times higher in OC patients compared to controls (MFI values 2602 (2296–2850) vs. 3894 (3196–4028), respectively, *p* < 0.001). Furthermore, it exhibited a tendency to decline toward control levels on day 5 postoperatively.

## 3. Discussion

### 3.1. Peripheral Blood Cell Counts, Ratios and Monocyte Cytokine Expression Are Altered in Preoperative OC Patients

OC significantly affects the tumor microenvironment, thereby impacting all cells within the tumor tissue, including immune cells [11,12]. Furthermore, the influence of OC extends beyond the tumor microenvironment to include various peripheral blood constituents, such as RBCs, WBCs, and Plts [43,44]. In our study, we observed higher concentrations of neutrophils and markedly elevated Plt counts in preoperative OC patients. These observed changes, when considering the inclusion and exclusion criteria, are likely attributable to the malignancy rather than infectious processes. Interestingly, Plts have gained significant attention in cancer progression investigations. Notably, higher Plt counts have been associated with a worse prognosis [45]. Conversely, elevated Plt counts have been linked to increased cancer incidences, including OC [46]. Studies on tumor-educated platelets (TEPs) also demonstrate promising results, suggesting their potential as a novel OC biomarker [47]. Subsequent investigation into PBMC subpopulation changes in preoperative OC patients revealed significant alterations, particularly in the proportions of CD8+ T cells and CD19+ B cells. Although CD19+ B cells are recognized for their regulatory role in tumors, with data supporting their prognostic value [13], their impact on anticancer immunity remains diverse and not fully understood [17,18]. In contrast, CD8+ T cells clearly contribute to anticancer immunity with their role in eliminating cancer cells established [11,48]. Therefore, a decreased proportion of CD8+ T cells in our study suggests impaired anticancer immunity. In the context of CD3+ T cells, which demonstrated an 8% decrease in OC patients, interpretation can be challenging. Firstly, existing literature predominantly emphasizes the significance of OC progression in relation to tumor-infiltrating CD3+ T cells [48,49]. Nevertheless, peripheral lymphocytes have also been shown to correlate with disease progression, with lower counts of peripheral lymphocytes being disadvantageous [50,51], independent of tumor-infiltrating lymphocytes [51]. Furthermore, the CD3+ T cell population is highly heterogeneous, with its significance lying in subsequent subsets. In addition to the aforementioned CD8+ T cells, other smaller subsets, such as tumor-promoting T regulatory lymphocytes and T helper 2 (Th2) lymphocytes, as well as tumor-inhibiting T helper 1 (Th1) lymphocytes, have a clearer impact on anticancer immunity [11,12]. Table 3 presents a brief overview of the findings within our study along with their potential implications on OC progression.

While not all PBMCs are universally recognized to correlate with disease prognosis, certain cell ratios may serve this purpose more effectively [23,24]. Upon the evaluation of lymphocyte-related ratios, we observed increased NLR, PLR, and CD4/CD8 ratios. While these were not primary outcomes, their alignment with the existing literature validates our findings [13,23]. Such alterations are under investigation for their prognostic value in disease progression or treatment success. Specifically, a high NLR and PLR have been demonstrated to correlate to poor PFS, OS, worse surgical outcomes, or poor response to chemotherapy [13,52,53]. Although less investigated, the CD4/CD8 ratio has attracted attention, with evidence suggesting that patients with a lower CD4/CD8 ratio, and therefore a higher proportion of CD8+ T cells, demonstrate better clinical outcomes [24,54]. Notably, we did not observe any significant changes in LMR, contrary to reports indicating its decrease in OC, which correlates with poor clinical outcomes [35,55]. Furthermore, a detailed correlation analysis was conducted between the investigated ratios, absolute counts of peripheral blood cells, and proportional values of PBMCs in OC patients and the control group. While these specific data are not included in the primary focus of this manuscript, they offer valuable insights that may contribute to further research into the immune response mechanisms observed in OC patients. 

The monocyte–macrophage cell lineage plays a pivotal role in the tumor microenvironment, representing key components of the innate immune response that are influenced by the adaptive immune system [11,12]. In cancer tissue, monocytes differentiate into tumor-associated macrophages (TAMs) [21,56], predominantly sourced from peripheral blood monocytes [57]. TAMs exhibit two distinct phenotypes, M1 and M2, determined by their cytokine environment, which in turn influences their impact on tumor progression. M1 macrophages are proinflammatory and suppress tumor growth, while M2 macrophages are anti-inflammatory and promote tumor growth [21,22]. Our findings show that OC patients exhibit a higher proportion of M2 monocytes before surgery along with a significantly lower M1/M2 monocyte ratio compared to controls. These results imply a state of immune suppression in OC patients, considering the characteristics of M2 monocytes. Our findings align with previous research indicating decreased PBMCs activity in OC patients, although there has been less focus on peripheral monocytes specifically [58,59,60]. Importantly, the majority of studies examining the role of the M1/M2 macrophages ratio in OC management have focused on macrophages within cancer tissue, with data showing that a high M1/M2 macrophage ratio is related to better OS and PFS rates [25,61,62]. The examination of the M1/M2 monocyte ratio in peripheral blood underscores the novelty of our study, highlighting the potential utility of peripheral monocyte ratios in future research.

It is important to note that different histological types of OC exhibit distinct immune profiles in both tumor tissue and peripheral blood [11,12,25]. While the distribution of histological types of OC patients in our study was comparable to that of the general population [41], the majority of cases were high-grade serous carcinoma. Therefore, our findings may be specific to this histology. When comparing the subgroups of high-grade serous carcinoma with other OC histological types in our study, we did not find any significant differences. Given the small sample size, these findings are most likely underpowered.

We further directed our analysis toward monocyte cytokine production to assess their activity in OC patients, as the monocyte/macrophage cell line is the most predominant line in OC tissue [22,23]. Cytokines play a pivotal role in immune cell communication, facilitating our understanding of immune system functions, albeit their effects can be complex to interpret due to their multifaceted nature [11,26]. Among the cytokines we investigated, IL-6 and IL-1β emerged as the most pro-tumorogenic [63,64,65]. Other cytokines, regardless of their proinflammatory (TNFα) or anti-inflammatory effects (IL-4, IL-10), may have diverse effects on promoting tumor growth [11,26,65,66,67,68,69]. In contrast, IL-12 acts as a key regulator of the immune response against tumors, promoting M1 macrophage polarization, the Th1 lymphocytes response, the activation of CD8+ T cells and NK cells, along with other mechanisms inhibiting tumor growth and progression [11,26,65]. Regarding our investigated cytokines, we did not find any statistically significant changes in IL-1β, IL-4, and IL-12 expression in OC patients’ monocytes. However, it is noteworthy that IL-4 is predominantly expressed by basophils, while IL-12 is mostly secreted by B lymphocytes and, to a lesser extent, by monocytes [70]. IL-10, known for its diverse actions but primarily recognized for its pro-tumorogenic potential [11,67], exhibited significant downregulation in M2 monocytes preoperatively. Additionally, TNFα displayed a tendency toward decreased expression in OC patients preoperatively. Despite TNFα’s acknowledged antitumor properties [66], current evidence supports its indirect involvement in pro-tumorogenic pathways [69,71,72]. Therefore, alterations in IL-10 and TNFα expression levels indicate a potentially less pro-tumorogenic environment. Furthermore, a noteworthy overexpression of IL-6 was observed in M2 monocytes of OC patients preoperatively. Considering IL-6’s established role in tumor progression [63,64,65], these variations suggest an intensified pro-tumorogenic environment in OC patients preoperatively. Summing up the cytokine expression data, it becomes evident that OC patients experience a disbalance in cytokine production. However, given the complex roles of the analyzed cytokines in inflammation and tumor progression, the final interpretation of our data remains ambiguous.

These findings from preoperative OC patients further highlight the systemic nature of OC, indicating that alterations in the immune system extend beyond the tumor microenvironment to involve the peripheral immune system. Despite the complexity in interpreting cytokine expression changes, the observed shifts in PBMCs in OC patients provide insights into the immunosuppressive environment in OC.

### 3.2. Early Postoperative Period Is Related to Additional Subtle Alterations in Peripheral Blood Cell Counts, Ratios and Monocyte Cytokine Expression in OC Patients

Surgery has been demonstrated to exert a significant impact on the immune system, with evidence suggesting that surgical interventions can impair immune function, potentially creating a window for the development of metastases [29,31,32]. Considering the significance of surgery in cancer patients, we further aimed to clarify the alterations occurring in PBMC subpopulations during the early postoperative period. Despite some insignificant fluctuations, our results suggest that surgery did not induce a subsequent decrease in CD3+ T cell and CD8+ T cell levels. Other authors similarly have not observed significant changes in CD3+ T and CD8+ T cell populations postoperatively [37,39]. Compared to controls, CD4+ T cells showed a significant increase 1 day postoperatively with CD19+ B cells demonstrating the most pronounced elevation. However, owing to the heterogeneous nature of CD4+ T cells [14,15] and inconclusive evidence regarding the impact of CD19+ B cells on cancer progression [17,18], definitive conclusions cannot be drawn from these findings alone. Nevertheless, other authors have reported beneficial shifts in smaller subsets of CD4+ T cell, particularly T regulatory cells, as early as day 2 after surgery, which persisted and became more pronounced 1 month after surgery [38,39]. It is noteworthy that another study reported opposite results 7 days postoperatively [37]. The postoperative analysis of lymphocyte-related ratios revealed minimal alterations, highlighting the potential utility of these ratios for prognosis in postoperative patients. This complements the preoperative use of LMR [35,55] and, to a lesser extent, the CD4/CD8 ratio [24,54].

Postoperative analysis of monocyte subsets indicated that M2 monocytes were most pronounced on the 1st day after surgery, although the M1/M2 ratio remained unchanged. However, investigations into peripheral blood monocytes following surgery are currently lacking. While not directly comparable to our findings, other studies have shown altered monocyte function after surgery with impaired peripheral monocyte function observed before surgery and persisting up to the seventh day after surgery [73,74]. Additionally, another study demonstrated the suppressed function of all PBMCs in the early postoperative period [40]. Although our study did not directly analyze monocyte activity, considering the immunosuppressive characteristics of M2 monocytes/macrophages [22] and the role of TAMs, particularly M2, in the metastatic progression of OC [21], our findings could have significant implications. Another intriguing discovery was the similarity in levels of M1 monocytes between controls and pre- and postoperative OC patients. This suggests that the change in M2 monocyte levels occurred at the expense of unclassified monocytes, whose dynamics in OC patients showed opposite trends compared to M2 monocytes. These findings support the plastic polarization theory, wherein existing subtypes may not be final and can change in response to environmental stimuli [21,75]. Furthermore, there is a growing interest in the role of monocytes in OC treatment [56]. In animal models of diverse cancer types, it has been demonstrated that the repolarization of monocytes from an M2 to M1 phenotype via immunotherapy can extend survival [76]. Consistent with the existing literature, our findings confirm that subsets of peripheral monocytes are indeed a subject for further investigation in OC patients during the early postoperative period.

In terms of cytokine expression in monocytes postoperatively, we observed only minor changes. Although we did not find significant alterations in cytokine expression postoperatively compared to preoperative levels, there was an increase in TNFα and IL-6 levels 1 day after surgery compared to controls. Additionally, IL-10 levels returned to levels similar to controls after being reduced before surgery. Thus, in addition to the preoperative cytokine disbalance, further subtle dysregulation in cytokine expression was noted. Given the known impact of these cytokines on OC progression [63,64,65,69,71,72], it could be speculated that the observed postoperative cytokine increase indicates a stronger immune suppression.

In summary, the postoperative analysis of PBMC subpopulations and monocyte cytokine expression revealed subtle changes. No strong correlations between OC surgical treatment and these changes were evident. While some fluctuations pose challenges in interpretation, others suggest that preoperative alterations in PBMCs and monocyte cytokine expression, indicative of immune suppression, persist after surgery and are not reversed during the early postoperative period. Conversely, considering the subtle postoperative changes, they may suggest an even more intensified pro-tumorigenic environment.

## 4. Materials and Methods

### 4.1. Study Design and Patient Selection

This prospective study enrolled 13 patients diagnosed with epithelial OC who underwent surgical treatment at the Hospital of Lithuanian University of Health Sciences Kauno Klinikos between January 2021 and April 2023. Surgical procedures included laparotomy and cytoreduction for OC patients without prior chemotherapy. The study group comprised patients with primary OC at stages III and IV according to the 2021 FIGO (International Federation of Gynecology and Obstetrics) staging system [42], excluding those with autoimmune diseases, other confirmed or suspected cancers, recent surgeries or blood transfusions within the past month. OC diagnosis was confirmed through histopathologic examination. A control group of 23 women matched in age and body mass index (BMI) with OC patients and free of cancer was included, adhering to the same exclusion criteria as the study group. Certain occurring non-malignant comorbidities, such as primary arterial hypertension, heart rhythm disorders, hypercholesterolemia, and obesity, were allowed in both the OC and control groups. Informed consent was obtained from all participants prior to enrolment, and the study was conducted in accordance with the Declaration of Helsinki principles and approved by the Kaunas Regional Biomedical Research Ethics Committee (Approval No. BE-2-16).

### 4.2. Blood Collection

Peripheral venous blood specimens were gathered from both OC patients and healthy controls. OC patient blood samples were acquired preoperatively and at 1 and 5 days postoperatively. Blood collection from healthy controls was a single-event procedure. A CBC analysis was conducted for both the control group and OC patients before surgery. Subsequent analyses were carried out for both groups, including blood sampling after surgery (Figure 6). Blood draws were conducted using either one 4 mL and one 10 mL vacutainer or a single 10 mL vacutainer containing K2EDTA (BD, Plymouth, UK). After blood sampling, the tubes were gently mixed to prevent clotting and promptly analyzed to minimize sample degradation.

### 4.3. Complete Blood Count Analysis

Venous blood samples collected in 4 mL K2EDTA-containing tubes were used for CBC analysis. An automated hematology analyzer UniCellDxH 800 (Beckman Coulter, Brea, CA, USA) was employed following the manufacturer’s instructions. The analyzer measured CBC parameters, including RBC count, Hb concentration, Plt count, WBC count, and differential leukocyte count.

### 4.4. Peripheral Blood Mononuclear Cells Isolation

Following blood collection in 10 mL K2EDTA-containing vacutainers, PBMCs were subsequently isolated using a Ficoll–Paque PREMIUM density gradient medium (Cytiva, Uppsala, Sweden) according to the manufacturer’s protocol. The isolated PBMCs were then carefully collected and subjected to two washes with PBS (Sigma-Aldrich, Steinheim, Germany) at 250× *g* for 5 min each. Once isolated from both OC patients and healthy participants, the PBMCs were resuspended to a concentration of 1 million cells per mL and divided into 1 mL aliquots for further analysis.

### 4.5. Flow Cytometric Analysis of Lymphocyte and Monocyte Subsets

To perform flow cytometric lymphocyte immunophenotyping, we used the BD multi-test 6-color TBNK kit (BD, Franklin Lakes, NJ, USA). The following subsets of lymphocytes were identified: CD3+ (T lymphocytes), CD3+/CD4+ (T helper lymphocytes), CD3+/CD8+ (cytotoxic T lymphocytes), CD19+ (B lymphocytes) and CD3−/CD16+56+ (NK cells). PBMCs aliquots containing 1 million cells each were incubated with a mixture of monoclonal antibodies (mAbs): CD3-FITC (Leu-4), CD4 PE-Cy™7 (Leu-3a), CD8 APC-Cy™7 (Leu-2a), CD16 PE (Leu-11c), CD19 APC (Leu-12), CD56 PE (Leu-19), and CD45 PerCP-Cy5.5 (2D1) according to manufacturer’s protocol. Following staining, the cells underwent washing steps and were analyzed using a 10-color flow cytometer, FACSLyric (BD, San Jose, CA, USA). Data acquisition was set to record up to 10,000 events per sample. Lymphocyte populations were gated based on granularity/complexity (side scatter—SSC) and CD45 expression, with the percentage of cells expressing specific antigens within the lymphocyte gate determined.

For the discrimination of monocytes/macrophages and their subsets, PBMCs were phenotypically analyzed using a panel of mAbs, including CD3 PE (UCHT1), CD14 BV510 (MφP9), CD16 APC (B73.1), CD80 APC-H7 (L307.4), CD86 PE-Cy7 (2331 (FUN-1)), CD163 BV605 (GHI/61), CD206 FITC (19.2), HLA-DR PerCP (L243), CD19 PE (HIB19), CD56 PE (555516) and CD66b PE (G10F5).

### 4.6. Flow Cytometric Analysis of Monocyte Cytokine Expression

For the assessment of cytokine production within monocytes, mAbs targeting IL-1β (H1b-98) Pacific Blue, IL-4 (MP4-25D2), IL-6 (MQ2-13A5), IL-10 (JES3-19F1), IL12 (C8.6), and TNFα (MAb11) (all purchased from BD, San Jose, CA, USA) were utilized. Intracellular cytokine accumulation was facilitated using the BD Cytofix/Cytoperm™ Plus Fixation/Permeabilization Kit with BD GolgiStop™ protein transport inhibitor containing monensin (BD, San Jose, CA, USA). Multicolor staining for cell surface antigens and intracellular cytokines was performed according to the manufacturer’s protocol provided with the kit. PMT voltage and compensation settings were adjusted using BD™ CompBeads Anti-Mouse Ig, κ/Negative Control Compensation Particles Set (BD, San Jose, CA, USA). A data acquisition goal of up to 30,000 total events per sample was set for monocyte/macrophage subsets and cytokine analysis. The M1 subset, characterized by CD45+/CD80+/CD86+ markers, and M2, identified by CD45+/CD163+/CD206+, were quantified as a percentage of cells within the lymphocyte gate. Cytokine expression levels were quantified as mean MFI values.

### 4.7. Statistical Analysis

Statistical analyses were conducted using SPSS (version 22.0; IBM Corp, Armonk, NY, USA) and GraphPad Prism software (version 9.5.1; GraphPad Software Inc., La Jolla, CA, USA). Given the sample size and non-normal distribution of most variables, we employed the Mann–Whitney test for independent variables and the Wilcoxon test for dependent variables comparison. Spearman’s correlation coefficient (rs) was used to examine correlations between variables. Quantitative results were presented as median with interquartile range (IQR), defined as quartile 1 to quartile 3, unless otherwise specified. Relative changes of cytokines were displayed after normalization to controls or preoperative OC patients (set at 1). Statistical significance was defined as *p* < 0.05.

## 5. Conclusions

### 5.1. Summary of Findings

Our investigation examines the immune dynamics in OC patients with a particular focus on PBMCs and cytokine production in the early postoperative period. Preoperatively, OC patients showed changes in PBMC subpopulations, with a decrease in CD8+ T cells and an increase in M2 monocytes, indicating systemic immune suppression that persisted after surgery. Cytokine analysis in OC patients showed an imbalance, with higher levels of tumor-promoting cytokines, notably IL-6, both before and after surgery, and increased levels of IL-10 and TNFα postoperatively. These findings highlight the systemic impact of OC and the potential consequences of surgery, showing that an impaired immune system and a tumor-promoting environment are present before surgery and accompany OC patients into the early postoperative period. Further research into postoperative immune dynamics in OC patients is needed, as the early postoperative period may offer opportunities for developing more targeted treatment strategies to improve OC outcomes.

### 5.2. Study Limitations

While our study provides valuable insights, several limitations need to be acknowledged. Primarily, the relatively small sample size may limit the generalizability of our findings, potentially characterizing our investigation as a pilot study. However, the novelty of certain experiments involving postoperative OC patients made it difficult to calculate an appropriate sample size. Additionally, our inclusion criteria restricted enrolment to stage III and IV OC patients according to the 2021 FIGO classification, which further limits the comparability of findings across different disease stages, consequently impacting generalizability. Furthermore, while we assessed changes in PBMC subpopulations and cytokine production, we did not investigate their implications for disease progression and clinical outcomes. Moreover, the absence of a control group with individuals undergoing benign gynecologic surgeries limits our ability to see the specific impact of OC on our results. Given the extensive nature of OC surgery, selecting appropriate control patients is inherently challenging. Lastly, the observational design of our study does not allow us to establish cause-and-effect relationships, requiring further research to understand the underlying mechanisms of immune changes in OC patients undergoing surgical treatment.

## Figures and Tables

**Figure 1 ijms-25-07087-f001:**
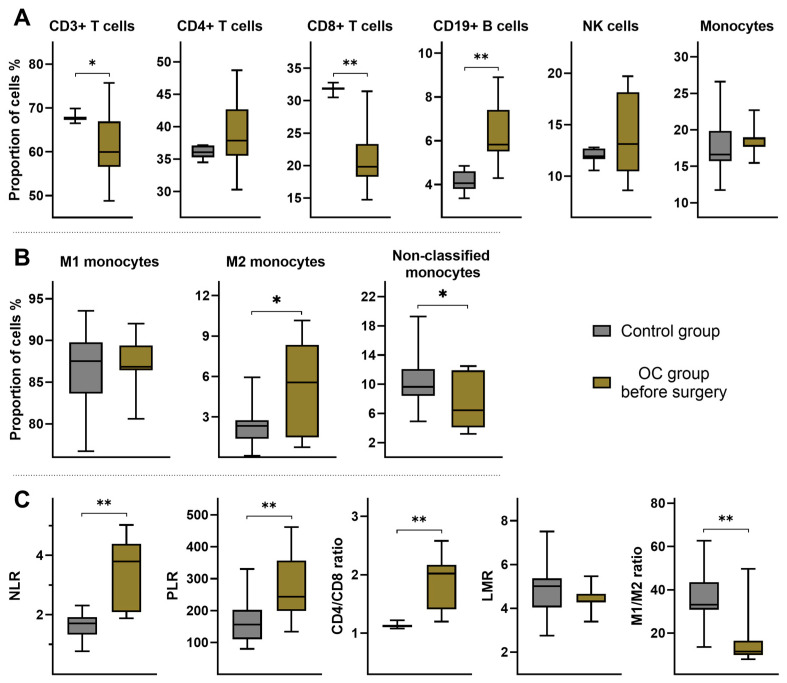
Changes in peripheral blood mononuclear cell (PBMC) subpopulations and ratios in preoperative OC patients. (**A**) The upper graphs illustrate the median proportions of T lymphocytes (CD3+ T cells), T helper lymphocytes (CD4+ T cells), cytotoxic T lymphocytes (CD8+ T cells), B lymphocytes (CD19+ B cells), natural killer cells (NK cells), and monocytes within all PBMCs. (**B**) The middle graphs illustrate the median proportions of monocyte subsets within all monocytes, including M1 (CD45+/CD80+/CD86+), M2 (CD45+/CD163+/CD206+), and non-classified monocytes (monocytes without a distinct phenotype). (**C**) The lower graphs depict median values of the neutrophil to lymphocyte ratio (NLR), platelet to lymphocyte ratio (PLR), CD4+ T cells to CD8+ T cells ratio (CD4/CD8 ratio), lymphocyte to monocyte ratio (LMR), and the M1 to M2 monocytes ratio (M1/M2 ratio). They represent data from the control group and OC patients before surgery. The boxes in the graphs represent the IQR, with the median indicated by a line inside the box. Error bars extend from the box to the minimum and maximum values of the dataset. All *p*-values between groups are >0.05 unless stated otherwise. * *p* < 0.05; ** *p* ≤ 0.001.

**Figure 2 ijms-25-07087-f002:**
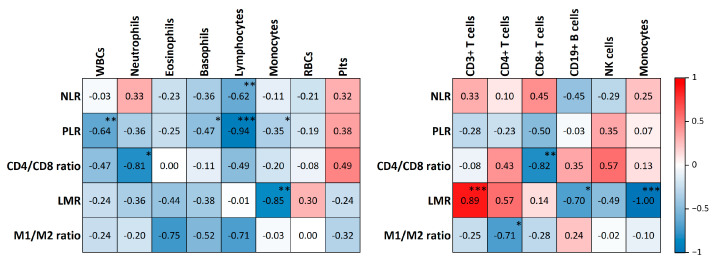
Correlation between peripheral blood cell ratios and peripheral blood cells in preoperative OC patients. The left graph presents the correlation between the ratios and quantitative values of blood cell counts. The right graph illustrates the correlation between the ratios and proportional values of PBMCs among all PBMCs. The listed values indicate Spearman’s correlation coefficient (rs). Colors in the graphs range from blue to red, indicating rs from −1 to 1, respectively. All *p*-values of correlations are >0.05 unless otherwise stated. * *p* < 0.05; ** *p* ≤ 0.01; *** *p* ≤ 0.001.

**Figure 3 ijms-25-07087-f003:**
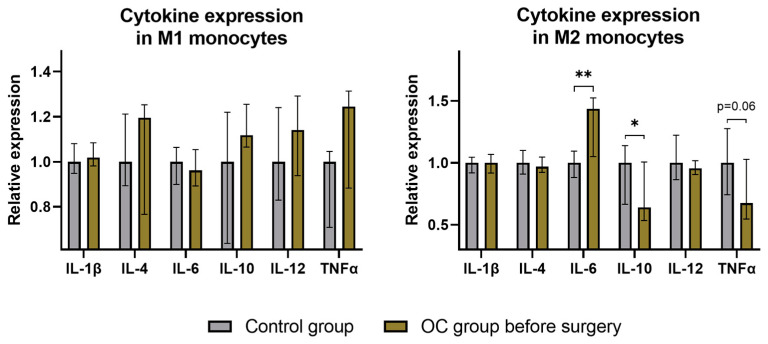
Alterations in cytokine expression in monocytes among preoperative OC patients. The graphs depict the fold change in cytokine expression, including interleukin-1 beta (IL-1β), interleukin-4 (IL-4), interleukin-6 (IL-6), interleukin-10 (IL-10), interleukin-12 (IL-12), and tumor necrosis factor alpha (TNFα), in M1 monocytes (left graph) and M2 monocytes (right graph). Columns represent median fluorescence intensity (MFI) values in OC patients before surgery, normalized to the control group, with error bars indicating the IQR. All *p*-values between groups are >0.05 unless stated otherwise. * *p* < 0.05; ** *p* ≤ 0.001.

**Figure 4 ijms-25-07087-f004:**
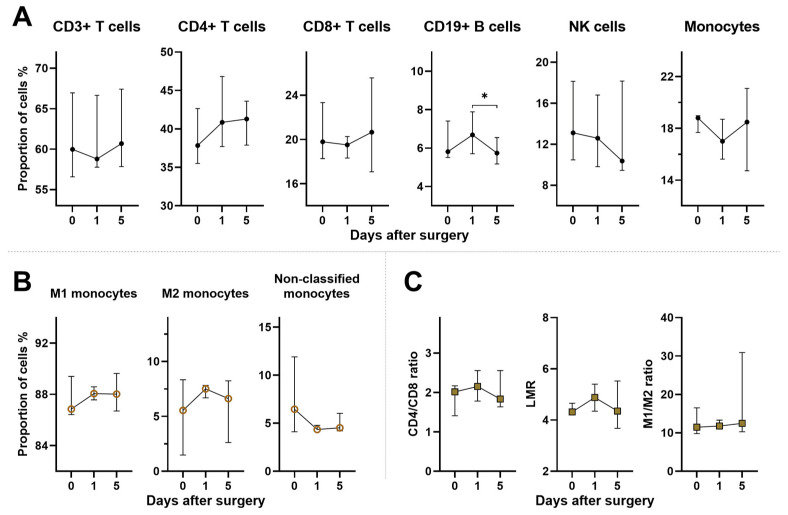
Changes in PBMC subpopulations and ratios in pre- and postoperative OC patients. (**A**) The upper graphs illustrate the median proportions of CD3+ T cells, CD4+ T cells, CD8+ T cells, CD19+ B cells, NK cells, and monocytes within all PBMCs. (**B**) The lower left graphs illustrate the median proportions of monocyte subsets within all monocytes, including M1, M2, and non-classified monocytes. (**C**) The lower right graphs depict median values of the CD4/CD8 ratio, LMR, and the M1/M2 ratio. The data represent OC patients before surgery (0) and on postoperative days 1 and 5. Graphs show median values with error bars indicating the IQR. All *p*-values between groups are >0.05 unless stated otherwise. * *p* < 0.05.

**Figure 5 ijms-25-07087-f005:**
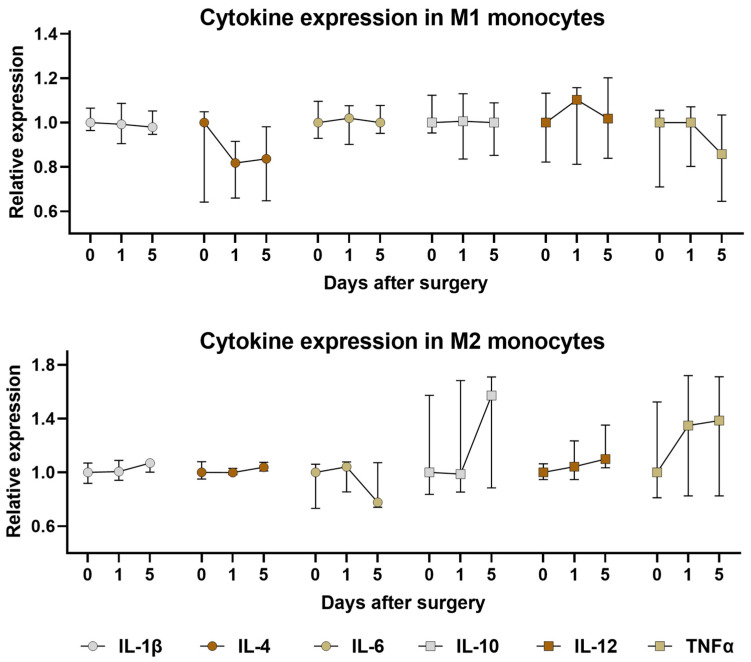
Alterations in cytokine expression in monocytes among pre- and postoperative OC patients. The graphs depict the fold change in cytokine expression (IL-1β, IL-4, IL-6, IL-10, IL-12, TNFα) in M1 monocytes (upper graph) and M2 monocytes (lower graph). Columns represent median MFI values in OC patients on days 1 and 5 after surgery, normalized to OC patients preoperatively (0), with error bars indicating the IQR. All *p*-values between groups are >0.05.

**Figure 6 ijms-25-07087-f006:**
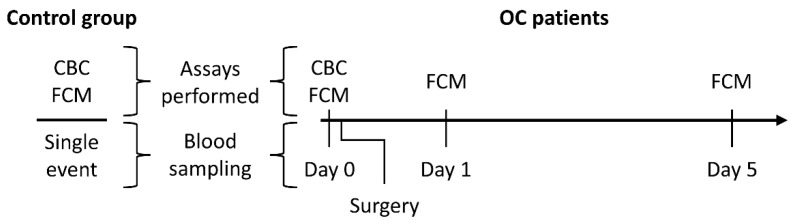
Blood sampling and assay schedule for study subjects. The lower section depicts blood collection events and their timing for the control group and OC patients. The upper section outlines the assays performed at each blood sampling: CBC, complete blood count; FCM, flow cytometry.

**Table 1 ijms-25-07087-t001:** Clinicopathological characteristics of the studied participants.

Characteristic	OC Group (*n* = 13)	Control Group (*n* = 23)	*p*-Value
Age (years), median (IQR)	58 (49–67)	59 (49–61)	0.75
Body mass index (kg/m^2^), median (IQR)	22.9 (22.1–28.8)	23.7 (21.9–28.7)	0.85
Stage of OC, n (%)			
IIIA	3 (23.1)	NA	NA
IIIB	3 (23.1)	NA	NA
IIIC	4 (30.7)	NA	NA
IVB	3 (23.1)	NA	NA
Histologic type of OC, n (%)			
Low-grade serous carcinoma	1 (7.7)	NA	NA
High-grade serous carcinoma	9 (69.2)	NA	NA
Mucinous carcinoma	1 (7.7)	NA	NA
Serous endometrioid carcinoma	2 (15.4)	NA	NA

OC, ovarian cancer; IQR, interquartile range; *n*, number of cases; NA, not applicable. The stages of OC are presented according to the 2021 FIGO (The International Federation of Gynecology and Obstetrics) staging system [42].

**Table 2 ijms-25-07087-t002:** Comparison of preoperative peripheral blood cell counts between the control group and OC patients.

Cell Type	OC Group (*n* = 13)	Control Group (*n* = 23)	*p*-Value
WBCs	6.8 (4.9–7.9)	5.6 (4.4–6.4)	0.09
Neutrophils	4.5 (3.3–5.3)	3.1 (2.3–3.5)	0.01
Eosinophils	0.1 (0.04–0.2)	0.1 (0.1–0.1)	0.36
Basophils	0.03 (0.03–0.06)	0.04 (0.02–0.04)	0.8
Lymphocytes	1.3 (1.1–1.8)	1.9 (1.3–2.3)	0.13
Monocytes	0.5 (0.2–1)	0.4 (0.4–0.5)	0.72
RBCs	4.2 (4–4.4)	4.5 (4.4–4.7)	0.06
Plts	372 (322–400)	267.5 (234–289)	<0.001

WBCs, white blood cells; RBCs, red blood cells; Plts, platelets. RBC counts are expressed in units of ×10^12^/L; other cell counts are expressed in units of ×10^9^/L. Cell counts are displayed as median values with interquartile ranges.

**Table 3 ijms-25-07087-t003:** Effects of PBMCs and cytokines on OC progression and main findings in preoperative OC patients.

Immune System Component	Effect on OC Progression	Current Study Findings in Preoperative OC Patients
CD3+ T cells	Complex	↓
CD4+ T cells	Complex	↔
CD8+ T cells	Tumor-inhibiting	↓
CD19+ B cells	Complex	↑
NK cells	Tumor-inhibiting	↔
M1 monocytes	Tumor-inhibiting	↔
M2 monocytes	Tumor-promoting	↑
IL-1β	Tumor-promoting	↔
IL-4	Complex	↔
IL-6	Tumor-promoting	↑
IL-10	Complex	↓
IL-12	Tumor-inhibiting	↔
TNFα	Complex	Tendecy of ↓

↑ increased level compared to controls; ↔ equal level compared to controls; ↓ decreased level compared to controls.

## Data Availability

The original contributions presented in the study are included in the article/Appendix A; further inquiries can be directed to the corresponding author.

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
