# Peer review of "Preoperative Immune Cell Dysregulation Accompanies Ovarian Cancer Patients into the Postoperative Period"

_ijms, 2024, doi:10.3390/ijms25137087_

Round 1

Reviewer 1 Report

Comments and Suggestions for Authors

The article by Ulevicius and colleagues examines changes in immune cells and cytokines in ovarian cancer patients before and after surgery. Firstly, this is an interesting topic and the article is very well written. Unfortunately, I regret to say that the manuscript is fundamentally flawed and I see no way to rescue it in its present form.

I am uncomfortable with figure 2. Firstly, the authors need to describe this analysis in more detail in the text. Secondly some of the observations seem self-evident – for example on the left of fig 2, if a patient has a large number of monocytes, it should be no surprise that lymphocyte to monocyte ratio is low. Similarly with lymphocytes and NLR.  This is also an issue in figure 1, where it should come as no surprise that if the number of CD4+ cells in OC patients is unchanged but the number of CD8+ cells decreases, the CD4/CD8 ratio goes up.  This data “over analysis”. Thirdly, there are a large number of correlations performed here. Where these planned before he study and was the study adequately powered to do all these analyses. Given the relatively low number of subjects in the study, I worry that there may be inadvertent p-hacking going on here?

Do I understand figure 4 and 5 correctly – the authors have compared PBMCs in ovarian cancer patients before and after surgery? If so, surely that is just telling us about the effects of surgery? To be fair to the authors, they do acknowledge this in the “limitations” section and I sympathize with the difficulty in finding appropriate controls, but unfortunately without this I don’t really see the value of this data.

I am very sorry to render a negative opinion, but I think the manuscript has serious isues.  All I can suggest is that the authors acknowledge these, and resubmit this as a short report with only report the data in Fig 1A 1B and 3  and say that this is a preliminary hypothesis generating study and a later study will seek to specifically examine the changes apparently observed here.

Author Response

Dear Sir or Madam,

We appreciate the invaluable insights and thoughtful observations you provided during the review of our manuscript. While we are disappointed to learn that you believe our work requires fundamental revisions, we would like to highlight that the manuscript has not yet been rejected by other reviewers. Consequently, we have diligently addressed each of the raised questions and points to enhance the coherence, accuracy, and overall strength of the manuscript.

Our response includes a detailed point-by-point address of the questions raised, with specific references to the manuscript for clarity. It is important to mention that in addition to addressing the highlighted sections based on your feedback, we have also made further revisions throughout the manuscript to accommodate inquiries from other reviewers.

Kind regards,

Jonas Ulevicius

Point-by-point response to Comments and Suggestions for Authors:

Comments 1: I am uncomfortable with figure 2. Firstly, the authors need to describe this analysis in more detail in the text. Secondly some of the observations seem self-evident – for example on the left of fig 2, if a patient has a large number of monocytes, it should be no surprise that lymphocyte to monocyte ratio is low. Similarly with lymphocytes and NLR.  This is also an issue in figure 1, where it should come as no surprise that if the number of CD4+ cells in OC patients is unchanged but the number of CD8+ cells decreases, the CD4/CD8 ratio goes up.  This data “over analysis”. Thirdly, there are a large number of correlations performed here. Where these planned before he study and was the study adequately powered to do all these analyses. Given the relatively low number of subjects in the study, I worry that there may be inadvertent p-hacking going on here?

Response 1: Thank you for pointing out the flaws regarding Figure 2. Although our correlation analysis is not the main focus of our manuscript, we believe it provides valuable insights for future research. We have added a brief explanation of the rationale for including these data in the “Discussion” section (pages 8-9, paragraph 3.1., lines 270-275). In response to your and other reviewers' comments, we have also edited the descriptions of Figures 2 and S1 (page 5, paragraph 2.2., lines 160-161).

We agree that some of the correlations presented in Figure 2 are quite obvious. However, we chose to retain these values in the correlation matrix for the sake of complete transparency in our calculations. Moreover, these obvious correlations, such as the CD4/CD8 ratio with CD8+ T cells, validate our findings.

Our study focuses on peripheral blood immune cells, thus, we presented not only the preoperative status of immune cells in OC patients but also immune cell ratios. These ratios are being investigated for clinical use by other researchers, and their importance is evolving. Our results align with findings from other studies, providing additional validation to our work. Presenting the alterations in investigated cells helps illustrate the origin of shifts in these ratios. As stated later in the text (page 8, paragraph 3.1., lines 261-263), presenting immune cell ratios was not our primary outcome. However, we believe that a detailed presentation of the gathered data might aid our understanding and further research into the immune response in OC patients.

Many of the correlations we performed are relatively novel, and as these calculations were not our primary outcome, underpowered results might be an issue. We acknowledge this in the “Study Limitations” section. However, we disagree that presenting these data constitutes "inadvertent p-hacking," despite some correlations being not readily explainable. This also does not imply that these findings are useless. Given the broad analysis of cell populations without in-depth subpopulation analysis, the meaning and origin of these findings could be clarified through more detailed future analysis. 

Comments 2: Do I understand figure 4 and 5 correctly – the authors have compared PBMCs in ovarian cancer patients before and after surgery? If so, surely that is just telling us about the effects of surgery? To be fair to the authors, they do acknowledge this in the “limitations” section and I sympathize with the difficulty in finding appropriate controls, but unfortunately without this I don’t really see the value of this data.

Response 2: Yes, by presenting Figures 4 and 5, we aimed to highlight the shift in immune cells and cytokines in response to surgery. Thank you for your valuable insight regarding this limitation of our study. As we discussed in the “Study Limitations” section, we considered incorporating an additional study group of patients with benign gynecological pathologies undergoing surgical treatment. However, we encountered challenges in finding benign pathologies that required surgical procedures similar in length and extent (total hysterectomy, omentectomy, peritoneal resection). Consequently, obtaining a suitable control group proved infeasible.

Introducing a control group with smaller surgeries and different anesthesia methods would likely introduce additional disparities, complicating result interpretation. Nonetheless, we cannot disregard the observed immune imbalances in post-surgical OC patients, even if they could be partly attributed to the healing process and other factors. We believe that, regardless of their origins, postoperative immune shifts are of great importance since surgery remains a core treatment for OC patients.

Comment 3: I am very sorry to render a negative opinion, but I think the manuscript has serious isues.  All I can suggest is that the authors acknowledge these, and resubmit this as a short report with only report the data in Fig 1A 1B and 3  and say that this is a preliminary hypothesis generating study and a later study will seek to specifically examine the changes apparently observed here.

Response 3: Thank you for your insights and suggestions. We agree that our study can be seen as a pilot study, and a more extensive investigation is needed. However, the study was planned, and we identified clear trends in how the immune system balance changes following surgical stress, suggesting that these findings may be clinically significant. These results provide a crucial foundation for further, possibly more specialized, targeted studies, including analysis of interventions aimed at reducing the negative impact of surgery. Therefore, we believe our manuscript should be published as it is. We have revised the “Study Limitations” section to incorporate your considerations (page 13, paragraph 6., line 501).

Reviewer 2 Report

Comments and Suggestions for Authors

In the submitted manuscript authors investigated peripheral blood mononuclear cells (PBMCs) subpopulations and cytokine production in monocytes of ovarian cancer (OC) patients both preoperatively and during the early postoperative period, and showed that preoperatively, OC patients exhibited changes in PBMC subpopulations, including decreased cytotoxic T cells, increased M2 monocytes, and disbalance of monocyte cytokine production. These alterations persisted after surgery, with subtle additional changes observed in PBMC subpopulations and cytokine expression in monocytes.

This manuscript is quite well written, study behind it is robust, and conclusion were corroborated with obtained results.

However, there are several things which should be further improved before this manuscript is suitable for publication:

1) In 'Abstract', studied PBMCs and cytokines should be precisely listed.

2) IQR should rather be presented as a range, not as an absolute value.

3) All abbreviations presented in figures and tables, like OC, FIGO, NA, PBMC subpopulations, etc., should be explained in respective legends and footnotes.

4) In figure legends it must be explained what those error bars on graphs present (I assume IQR).

5) In the main text actual p-values should always be provided after stating the results of statistical comparisons.

6) Figure S1 legend must contain all the same data as for Figure 2, while in both it would be useful to state that Spearman's correlation coefficient was calculated.

7) The assessment of cytokine production should be put in separate subsection of the 'Materials and Methods' section.

8) For the sake of reproducibility, it is suggested that authors provide all per sample results in the supplementary table.

9) The biggest drawback of this manuscript is that authors did not take into serious consideration the existence of histological sub-types of OC. Although it is obvious that authors' OC sample size was too small for sub-group analyzes, separate statistical analyses could be performed for HGSOC patients. However, authors should definitively tackle this issue in 'Discussion', and at least speculate if their observed results are pan-OC or could be subtype-specific!

Author Response

Dear Sir or Madam,

We would like to express our sincere gratitude for the invaluable insights and thoughtful observations provided during the review of our manuscript. Your feedback has been instrumental in refining the quality of our work, and we truly appreciate the time and effort you dedicated to offering such constructive comments. We have diligently addressed each of the raised questions and points to enhance the coherence, accuracy, and overall strength of the manuscript.

Our response includes a detailed point-by-point address of the questions raised, with specific references to the manuscript for clarity. It is important to mention that in addition to addressing the highlighted sections based on your feedback, we have also made further revisions throughout the manuscript to accommodate inquiries from other reviewers.

Kind regards,

Jonas Ulevicius

Point-by-point response to Comments and Suggestions for Authors:

Comments 1: In 'Abstract', studied PBMCs and cytokines should be precisely listed.

Response 1: Thank you for pointing this out. We have added a list of PBMCs and cytokines to the abstract (page 1, lines 17-18).

Comments 2: IQR should rather be presented as a range, not as an absolute value.

Response 2: Thank you for your comment. From a statistical perspective, the interquartile range (IQR) is typically presented as a single value. However, it is sometimes shown as the range between the first and third quartiles (Q1-Q3). This latter presentation provides more detailed information about data dispersion. We agree with your suggestion and have edited all IQR values accordingly (pages 3-7, paragraphs 2.1, 2.2, 2.3). We have also edited the “Methods and materials” section (page 12, paragraph 4.7, line 482).

Comments 3: All abbreviations presented in figures and tables, like OC, FIGO, NA, PBMC subpopulations, etc., should be explained in respective legends and footnotes.

Response 3: By not explaining every abbreviation in figures and tables we followed the journal's recommended standart: “Acronyms/Abbreviations/Initialisms should be defined the first time they appear in each of three sections: the abstract; the main text; the first figure or table”. However we agree that some abbreviations were left unexplained even though they were mentioned for the first time. We have edited the text, tables and figures accordingly (page 3, Paragraph 2.1., lines 109-111; page 11, paragraph 4.1, lines 402-403).

Comments 4: In figure legends it must be explained what those error bars on graphs present (I assume IQR).

Response 4:  We have added the missing information to the figure legends as follows: page 5, paragraph 2.2., line 152-154, 173-174; page 6, paragraph 2.3., line 199; page 7, paragraph 2.3., line 221).

Comments 5: In the main text actual p-values should always be provided after stating the results of statistical comparisons.

Response 5: Thank you for pointing this out. We have edited the positions of p-values in the text (page 6, paragraph 2.3., lines 184, 185, 187, 206).

Comments 6: Figure S1 legend must contain all the same data as for Figure 2, while in both it would be useful to state that Spearman's correlation coefficient was calculated.

Response 6: We have equalised the legends and specified that Spearman’s correlation was calculated (page 5, paragraph 2.2., lines 160-161).

Comments 7: The assessment of cytokine production should be put in separate subsection of the 'Materials and Methods' section.

Response 7: Thank you for the suggestion. We have now separated the section about the assessment of cytokine production in the "Materials and Methods" (page 12, paragraph 4.6., line 461).

Comments 8: For the sake of reproducibility, it is suggested that authors provide all per sample results in the supplementary table.

Response 8: Thank you for the suggestion. We believe the manuscript already contains a substantial amount of descriptive statistics and basic calculations, which are straightforward and easy to understand. Including raw data in the supplementary files would unnecessarily overload the manuscript. However, we have selected the "Dataset available on request from the authors" option provided by the journal, and we will share the raw data upon request.

Comments 9: The biggest drawback of this manuscript is that authors did not take into serious consideration the existence of histological sub-types of OC. Although it is obvious that authors' OC sample size was too small for sub-group analyzes, separate statistical analyses could be performed for HGSOC patients. However, authors should definitively tackle this issue in 'Discussion', and at least speculate if their observed results are pan-OC or could be subtype-specific!

Response 9: Thank you for this important note. We have incorporated your comment and provided further discussion in the "Discussion" section (page 9, paragraph 3.1., lines 295-302).

Reviewer 3 Report

Comments and Suggestions for Authors

Dear Authors,

A lot of your paper seems written by AI. While I personally have nothing against this practice, I do discourage the use of AI without further adjustment of the generated text because it is too complex and hard to follow. So please rewrite the text so that it is more readable and easier to understand.

You can find my comments in the pdf.

Comments on the Quality of English Language

a lot of the manuscript should be rewritten due to complex language that seems AI generated

Author Response

Dear Sir or Madam,

We would like to express our sincere gratitude for the invaluable insights and thoughtful observations provided during the review of our manuscript. Your feedback has been instrumental in refining the quality of our work, and we truly appreciate the time and effort you dedicated to offering such constructive comments. We have diligently addressed each of the raised questions and points to enhance the coherence, accuracy, and overall strength of the manuscript.

Our response includes a detailed point-by-point address of the questions raised, with specific references to the manuscript for clarity. We extracted all the comments from the PDF file and provided context for each comment to ensure better understanding. Additionally, we have made further revisions throughout the manuscript to address inquiries from other reviewers, beyond the highlighted sections based on your feedback.

Kind regards,

Jonas Ulevicius

Response to Comments on the Quality of English Language:

(comments and marked text for correction were presented in the attached PDF file)

Comments 1: A lot of your paper seems written by AI. While I personally have nothing against this practice, I do discourage the use of AI without further adjustment of the generated text because it is too complex and hard to follow. So please rewrite the text so that it is more readable and easier to understand.

Response 1: We have made the required text modifications for better coherence (page 1, abstract, lines 25-27; pages 7-10, paragraph 3.1., lines 225, 227, 232, 234, 235, 241, 253, 281, 303, 323; 328-330; pages 10-11, paragraph 3.2., lines 341; page 13, paragraph 6).

Comments 2: (about paragraph 3.2., lines 357-386) “too complex. please simplify”.

Response 2: We have made multiple changes in paragraph 3.2. for better coherence (pages 10-11, paragraph 3.2., lines 358-386).

Point-by-point response to (other) Comments and Suggestions for Authors:

(comments and marked text for correction were presented in the attached PDF file)

Comments 3: (for the “Results” section) “I suggest adding some analyses with the clinical features of the disease. CA125, PCI, Aletti score, CC score to name a few”.

Response 3: Thank you for your suggestion. We agree that the analysis of coexisting clinicopathological features is crucial in understanding the immune system's response to ovarian cancer and surgery. However, these evaluations fall beyond the scope of our current investigation. Nonetheless, these additional analyses could be a future aim with a larger study sample, incorporating even more important aspects, such as survival analysis.

Comments 4: (about the control group)- “further details are needed. Are they completely healthy controls?”.

Response 4: Thank you for your comment. Although the control group participants did not have any malignancies or multiple severe comorbidities, some had common non-malignant conditions. Given that the study group had similar comorbidities, this similarity can be seen as beneficial for creating a more matched control group. We have edited the “Materials and methods” section for better coherence (page 11, paragraph 4.1, lines 408-410).

Comments 5: (about Table 1) “add this next to the variable instead of making notes with * **”.

Response 5: We have made the required changes (page 3, paragraph 2.1., line 108).

Comments 6: (about Figure 1) “please use boxplots instead as they are more appropriate”.

Response 6: We have edited the figure (page 4, paragraph 2.2, line 142).

Comments 7: (about Figure 3) “I also suggest choosing another color that is more eyepleasing.”

Response 7: We have changed the colours in Figure 3 (page 5, paragraph 2.2., line 169).

Comments 8: (question about the control group) “did patients have other comorbidities?”.

Response 8: Thank you for pointing this out. We have specified the comorbidities while answering the 4th comment. See page 11, paragraph 4.1, lines 408-410.

Comments 9: (about the control group participants having cancer in their past) “even in their past?”.

Comment 9: Yes, we chose not to include participants with any confirmed or suspected cancers, even those diagnosed in the past.

Comments 10: (about “Conclusions” section) “too long. state only the most relevant findings”.

Response 10: We have adited and shortened the “Conclusions” section (page 13, paragraph 5, lines 486-497).

Comments 11: (about OC group patients)“did any of the patients receive transfusion during surgery? PDS is usually aggressive in advances stages. What was the tumor residue after PDS?”.

Response 11: Thank you for the question. While such patients do sometimes require blood transfusions during treatment, none of our participants received blood transfusions. Regarding the extent of cytoreduction: 8 (61.5%) of the surgeries were radical, 4 (30.8%) were optimal, and 1 (7.7%) was suboptimal.

Round 2

Reviewer 1 Report

Comments and Suggestions for Authors

I am genuinely sorry but I remain unconvinced by the authors responses to my previous  questions. 

The authors' response to my question about "inadvertent p-hacking" was to essentially agree with me. Indeed they then say that further studies are necessary to confirm the observation. I agree completely and for that reason I proposed, and still think, that the best route forwards for them is to rewrite this manuscript to make sure it is clear that this as a pilot study generating hypotheses. I agree completely that the observations are not useless (their words, not mine). They are a starting point for a subsequent study.

The same applies to the changes in cytokines after surgery . 13 patients were examined. In figure 4 alone, I think 36 different comparisons were performed. 

Reviewer 3 Report

Comments and Suggestions for Authors

Regarding the extent of cytoreduction: 8 (61.5%) of the surgeries were radical, 4 (30.8%) were optimal, and 1 (7.7%) was suboptimal. - please add this information in Table 1.

Please add "a pilot study" in the title. You have very few patients.

Comments on the Quality of English Language

The text has been improved.